# Analysis of HLA Alleles in Different Cohorts of Patients Infected by *L. infantum* from Southern Spain

**DOI:** 10.3390/ijms25158205

**Published:** 2024-07-27

**Authors:** Juan Francisco Gutiérrez-Bautista, Antonio Sampedro, Lucia Ballesta-Alcaraz, María Aguilera-Franco, María José Olivares-Durán, Fernando Cobo, Juan Antonio Reguera, Javier Rodríguez-Granger, Andrés Torres-Llamas, Joaquina Martín-Sánchez, Inés Aznar-Peralta, Jose Ramon Vilchez, Miguel Ángel López-Nevot, Antonio Sampedro-Martínez

**Affiliations:** 1Departamento de Bioquímica, Biología Molecular e Inmunología III, University of Granada, 18016 Granada, Spain; 2Servicio de Análisis Clínicos e Inmunología, University Hospital Virgen de las Nieves, 18014 Granada, Spain; luciaballesta2103@gmail.com (L.B.-A.); mjolivaresduran@gmail.com (M.J.O.-D.); joser.vilchez.sspa@juntadeandalucia.es (J.R.V.); 3Instituto de Investigación Biosanitaria de Granada (ibs.GRANADA), 18012 Granada, Spain; antonioj.sampedro.sspa@juntadeandalucia.es; 4Centro de Salud Zaidín Sur, Distrito Granada Metropolitano, Servicio Andaluz de Salud, 18007 Granada, Spain; antonio.sampedro.padilla@gmail.com; 5Servicio de Microbiología, University Hospital Virgen de las Nieves, 18014 Granada, Spain; maguilfran@gmail.com (M.A.-F.); fernando.cobo.sspa@juntadeandalucia.es (F.C.); jantonio.reguera.sspa@juntadeandalucia.es (J.A.R.); granger022@gmail.com (J.R.-G.); 6Departamento de Parasitología, University of Granada, 18016 Granada, Spain; andrestll@ugr.es (A.T.-L.); joaquina@ugr.es (J.M.-S.); 7GENYO Centre for Genomics and Oncological Research, Pfizer, University of Granada, Andalusian, 18016 Granada, Spain

**Keywords:** leishmaniasis, human leukocyte antigen (HLA), *L. Infantum*, haplotype, Immunogenetics

## Abstract

Leishmaniasis is an infectious disease caused by protozoa of the genus *Leishmania*, which is endemic in certain areas of Europe, such as southern Spain. The disease manifests in various clinical phenotypes, including visceral, cutaneous, mucosal, or asymptomatic leishmaniasis. This diversity in clinical outcomes may be influenced by the host immune response, with human leukocyte antigen (HLA) molecules playing a crucial role in determining susceptibility and progression of the infection. This study explores the association between specific HLA variants and *Leishmania infantum* infection. We recruited four cohorts: a control group, asymptomatic individuals, patients with symptomatic disease, and cohabitants of infected individuals. HLA typing was performed for all participants, followed by an association analysis with infection status and disease progression. Our findings indicate that the HLA-B*38 and HLA-C*03 alleles are associated with protection against *L. infantum* infection. These results contribute to a better understanding of the disease’s progression, offer potential for new therapeutic approaches such as vaccines, and expand the existing knowledge in the literature.

## 1. Introduction

Leishmaniasis is a zoonotic disease that has a high incidence worldwide, being present in more than 98 countries [1]. In terms of epidemiology, poor countries are the most affected. The number of leishmania cases has been estimated to be between 0.2 and 0.4 million cases worldwide, with 90% reported in India [2]. Mortality is estimated at 10–20% and is centered in the poorest countries [2]. The incidence in Europe is low, with an average of 700 cases reported annually across multiple countries [3]. The prevalence in Spain is reported to be 0.3 cases per 100,000 inhabitants, with the Mediterranean basin in southern Spain being an endemic area for *Leishmania infantum* [4]. The cumulative incidence in Spain from 2005 to 2020 is 0.67 cases per 100,000 inhabitants [5].

The infection is caused by protozoa of the genus Leishmania. The clinical spectrum of the disease is broad, with up to six different types of clinical forms: visceral (VL), post-kala-azar dermal leishmaniasis (PKDL), cutaneous (CL), diffuse cutaneous (DCL), mucocutaneous (MCL) and mucosal (ML) leishmaniasis [1]. However, most patients do not develop any symptoms (asymptomatic patients) [6]. The main vectors in Spain are three species of phlebotomine sandflies of the subgenus *Larroussius* (*Phlebotomus perniciosus*, *Phlebotomus ariasi,* and *Phlebotomus langeroni*) [7]. In human infection, anthroponotic transmission occurs, with the main reservoir being the dog, resulting in a domestic cycle [2].

Genes and polymorphisms associated with the development of symptomatic or asymptomatic disease, as well as risk or protection against leishmaniasis infection, have been identified [8]. Among these proteins, the human leukocyte antigen (HLA) molecules are of great interest, which are associated with infections by different pathogens such as SARS-CoV-2, hepatitis C virus (HCV), influenza virus, human immunodeficiency virus (HIV) and plasmodium [9,10,11]. These molecules belong to the major histocompatibility complex (MHC) and are the most polymorphic molecules in humans [12]. Their function is essential for the normal development of the adaptive immune response to infections. They are responsible for the presentation of pathogen-derived peptides to T cells to orchestrate the immune response [13]. Therefore, the presence of certain alleles may or may not favor the development of the immune response and may explain the different clinical forms and conditions of the patient’s evolution. In infections by intracellular microorganisms, HLA class I (HLA-I) alleles are of major importance. In the elimination of intracellular pathogens, cell-mediated immunity is crucial, where cytotoxic CD8 T cells recognize the antigen in the context of an HLA-I molecule. In the case of *L. infantum* infection, HLA-I molecules will express protozoan-derived proteins in the membrane. Alleles that have a higher affinity for *L. infantum* peptides or that bind more immunogenic peptides will induce better cytotoxic responses that will better eliminate infected cells.

HLA presents a series of epitopes involved in the effector functions of innate immunity. HLA-B alleles present a polymorphism in the α-1 domain (residues 77 to 83) that divides all HLA-B alleles into two groups: those presenting the Bw4 epitope (some HLA-A alleles present it) and those expressing Bw6 [14]. Their importance resides in the activation/inhibition of natural killer (Nk) cells, which recognize these epitopes through their killer immunoglobulin-like receptors (KIR) [15]. HLA-I molecules are the most crucial ligands for KIRs; the development and response of NK cells are highly dependent on the KIR/HLA-I interaction. For example, the Bw4 epitope is an inhibitor for the KIR3DL1 allele, whereas it is an activator for the KIR3DS1 receptor [15]. On the other hand, HLA-C alleles are classified into two groups, C1 and C2. The C1 group has asparagine at position 80 (Asn80), and the C2 group has lysine at position 80 (Lys 80) [15]. This ability to discriminate and modulate KIRs according to HLA epitopes underscores the complexity and specificity of the innate immune system in its recognition and response to various stimuli, which further enhances the value of HLA molecules.

To determine the role of HLA alleles in *L. infantum* infection, we performed the first descriptive analysis of Andalusia, southern Spain, with four different cohorts: (i) a cohort of patients with asymptomatic leishmania infection; (ii) a cohort of patients with visceral, cutaneous or mucosal infection; (iii) a cohort of cohabitants of people with leishmania disease; (iv) a representative control group of the area. We investigated the impact of HLA in the development of symptomatic or asymptomatic infections and their protective or risk role in *L. infantum* infection. For this purpose, HLA molecular typing was performed in individuals of each group, analyzing allelic and haplotypic frequencies, homozygosity, Bw4/Bw6 epitopes, and C1/C2 groups. In short, the main objectives of this study were to analyze the association of HLA alleles with susceptibility and progression of *L. infantum* infection and to identify potential protective or risk HLA alleles.

## 2. Results

### 2.1. Demographic Data

Sex and age were compared between the different study groups. No significant differences were detected for sex (*p* = 0.295) or age (*p* = 0.167) between the groups studied.

### 2.2. HLA Typing

The HLA typing of the individuals composing each group can be found in Appendix A. They were used to look for differences in allelic and haplotypic frequencies and homozygosity, as well as in the frequency of Bw4/Bw6 epitopes and C1/C2 groups.

### 2.3. Allele Frequencies

The different frequencies of each allele in the different cohorts were analyzed (Appendix A).

In the class I region, there were 20, 26, and 14 alleles at the HLA-A, B, and Cw loci, respectively. In the class II region, there were 14 alleles at the HLA-DRB1 locus and 5 alleles at the HLA-DQB1 locus. HLA-A*24 was associated with infection, HLA-A*34 was increased in the control group against symptomatic disease, HLA-B*07, B*15, B*38, B*, C*03, and DR*11 were increased in the cohabitants group against controls, and HLA-B*44 was increased in the control group against cohabitants group. Nevertheless, when the *p* value was corrected (Bonferroni’s correction), only the *p* values for HLA-B*38 and C*03 were statistically significant (Table 1).

To further study the association between HLA alleles that remained significant after Bonferroni correction, we performed a logistic regression analysis. Specifically, we analyzed the association of HLA-B*38 and HLA-C*03 alleles with infection. The analyses showed an inverse association with infection. The HLA-B*38 allele presented a *p* value of 0.071 with an Odds Ratio (OR) of 0.317 (95% CI: 0.120–0.834), and the HLA-C*03 allele had a *p* value of 0.02 with an OR of 0.326 (95% CI: 0.096–1.100).

### 2.4. Haplotypes

An analysis of the extended haplotypes in each of the cohorts was performed (Appendix A). Of the haplotypes present, only haplotypes 18.3 (HLA-A*02:01, B*18:01, C*05:01, DRB1*11:02, DQB1*03:01) and 60.3 (HLA-A*02:01, B*40:01. C*03:01, DRB1*13:02, DQB1*06:04), were increased in LD and cohabitants, respectively, versus the control group. However, *p* values were not significant after the Bonferroni correction (Table 2).

### 2.5. Homozygosis

We determined the frequency of homozygosity for HLA-A, -B, -C, -DRB1, and -DQB1 loci. We found no significant differences between the groups (Table 3).

### 2.6. Bw4/Bw6 Epitopes and C1/C2 Groups

We determined the frequency of heterozygosis for the Bw4-Bw6 epitopes and of homozygosis for Bw4 and Bw6 (Table 4). Finally, we analyzed the frequency of heterozygotes and homozygotes for the C1 and C2 groups. We detected no significant differences between the groups (Table 4).

## 3. Discussion

Leishmania infection generates a wide range of clinical phenotypes. Infection is mainly asymptomatic, but in cases where the immune system is depressed, as in acquired immunodeficiency syndrome (AIDS) or primary immunodeficiency (PID), infection can be lethal. However, the development of the disease is the result of several variables, including host-parasite interaction, immune system status, and genetic factors [16]. The World Health Organization (WHO) recognizes leishmaniasis, including infection caused by *L. infantum*, as a significant public health problem [17]. WHO’s efforts to control the disease are numerous. They have recently developed a program to eradicate visceral leishmaniasis in East Africa [18]. Therefore, it is important to understand the pathogen-host interaction and the role of genetic factors in disease susceptibility and resistance. Among the genetic factors, HLA molecules play a fundamental role in the adaptive immune response. Knowledge of the association between specific HLA variants and Leishmaniasis has important therapeutic implications. It allows the development of personalized medicine, designing more specific and effective treatments according to the patient’s HLA variants. It facilitates the creation of more effective vaccines, guided by the understanding of how certain variants influence the immune response. Improves disease prognoses by identifying variants associated with more severe forms, allowing earlier and more aggressive interventions. It drives the development of new immunotherapies by understanding the interactions between HLA, the immune system, and the parasite. In addition, it helps to design public health strategies adapted to the genetic profile of populations, optimizing the prevention and control of Leishmaniasis. Thus, we analyzed HLA allele and haplotypic frequencies to identify possible associations with disease severity and susceptibility to or protection from infection. Among other associations, associations have been reported in HLA-I alleles with different parasite infections such as giardiasis, malaria, and schistosomiasis [10,19,20,21].

Comparisons with the different groups showed values close to significance for HLA-A*24, -A*34, -B*07, -B*15, -B*38, -B*44, -C*03, and -DRB1*11 alleles. However, only HLA-B*38 and HLA-C*03 alleles showed significant values after correction by multiple testing. These alleles were increased in the cohabitant group compared to the control group, so they could be protection-associated alleles. In addition, both alleles were more represented in the cohabitant and asymptomatic groups, while the lowest frequency was found in the control and symptomatic groups. This supports that HLA-B*38 and HLA-C*03 alleles play a protective role in *L. infantum* infection, both in terms of protection against infection and in the development of asymptomatic disease. The HLA-B*38 allele has been associated with post-transplant lymphoproliferative disorder in solid-organ transplant recipients or susceptibility to develop psoriatic arthritis in the Argentine population [22,23].

Studies have shown an association of the HLA-B*15 allele with protection from infection [24], while others associate it with susceptibility to infection [25]. In our results, despite not showing significant values after Bonferroni correction, it tends to be associated with a protective role since it is more frequent in the groups of cohabitants and asymptomatic patients, which coincides with the results of Lara M. et al. [24].

The HLA-B*07 allele was found to be overrepresented in controls in the work of Samaranayake N. et al., whereas our results show a decrease in frequency in the control group with an increase in infected populations. [25].

Regarding HLA-II molecules, we report a possible protective role of the HLA-DRB1*11 allele since it is increased in the cohort of cohabitants. However, some authors associate it with risk [26,27]. Other class II alleles associated with Leishmania are HLA-DRB1*15:01—associated with protection [8] or risk [25]—and HLA-DRB1*14:04 and DRB1*13:01 alleles—as risk factors [8].

It is important to highlight the wide variety of contradictory results found in the literature. This may be due to the different ethnicities studied, the species/strain producing the infection, and the different HLA frequencies found in each geographical area studied.

Our results indicate a predominant role of HLA-I versus HLA-II molecules in *L. infantum* infection. HLA-I molecules are responsible for the antigenic presentation of intracellular proteins, such as those derived from intracellular pathogens. Better presentation of peptides derived from *L. infantum* proteins will lead to better disease control by CD8+ T cells. Conversely, HLA molecules that do not effectively present protozoan epitopes will lead to poor infection control and the development of a more severe infection.

Regarding the haplotypes appearing in our different cohorts, haplotypes 18.3 and 60.3 were close to significant. Haplotype 18.3 was overrepresented in patients with LV versus the control group, which seems to indicate risk. In this case, we found that HLA-DRB1*11:02 and DQB1*03:02 alleles have been reported as possible risk alleles, although never in haplotype form, which reinforces our results [26,27,28].

On the other hand, haplotype 60.3 was increased in the group of cohabitants, which may indicate a protective role. This haplotype contains the HLA-C*03 allele, which is associated with protection in our cohort, but there are no previously published results on this allele. Regarding HLA-C alleles, HLA-C*04 was associated with a risk of *L. braziliensis* infection [29]. Concerning the HLA-II alleles that are part of the haplotype, it has been published that the HLA-DRB1*13 allele is associated with a risk of infection [26,27].

In the analysis of homozygosity for HLA alleles, we did not obtain significant differences between the different cohorts. Homozygosity at HLA loci produces a decrease in the number of peptides that can be presented to T lymphocytes, theoretically decreasing the potential for adaptive immunity. Among the diseases found to be related to HLA homozygosity are COVID-19 and increased risk of progression in HIV-1 infected individuals [30,31].

Finally, we performed an analysis of Bw4/Bw6 epitopes and C1/C2 groups. These epitopes are ligands for Nk lymphocyte activation/inhibition proteins called KIR [32]. These lymphocytes are part of the innate immune response and, therefore, can be of great help in the early stages of infection. The KIR alleles interact with the different HLA allele groups, including Bw4/Bw6 and the C1 and C2 groups. On the one hand, KIR2DL1, KIR2DL2, and KIR2DL3 receptors can discriminate between C1 and C2 groups, allowing inhibition of NK cells [33]. In contrast, KIR2DS1 and KIR2DS2 receptors recognize the C2 and C1 groups, respectively, generating an activating response. While KIR3DL1 and KIR3DS1 receptors are able to directly recognize the Bw4 epitope, producing inhibition and activation, respectively [32], our results did not show significant results. Nevertheless, it is important to study this type of polymorphism since they have been associated with several diseases [34]. For example, the presence of Bw4 has been associated with delayed development of AIDS in HIV patients [33].

In conclusion, we present the results of an analysis of HLA frequencies and haplotypes in different cohorts of Leishmania patients from the Spanish Mediterranean account. We show a possible association between HLA-B*38 and HLA-C*03 alleles and protection against infection, which should be confirmed with larger cohorts to obtain greater statistical power. The remaining analyses, homozygosity, Bw4/Bw6 epitopes, and C1/C2 groups, did not show significant results. Despite these results, it is important to point out that our study has several limitations, the most important being the small sample sizes of asymptomatic infection, LD, and cohabiting groups, which likely result in small statistical power of the study.

## 4. Materials and Methods

### 4.1. Study Groups

The patients in all groups were inhabitants of the province of Granada, Andalusia, in southern Spain. Demographic data are presented in Table 5. Ethics approval was granted by the Human Research Ethics Committee. After informed consent was obtained from each participant, blood samples were drawn for leishmanial serology and real-time polymerase chain reaction (rt-PCR). All samples were obtained between 2015 and 2022.

### 4.2. Control Group

The control group (*n* = 636) consists of healthy blood donors who are representative of the Granada area [35]. Healthy individuals who did not present any disease of autoimmune etiology or immunodeficiencies were selected. The average age of the group is 45 (range 18–65 years), with 51% of its members being women (*n* = 324).

### 4.3. Asymptomatic Infection

This group was composed of 55 individuals, wherein 23 donors were female (41,8%), and the average age was 41 (range 18–65 years). For this study, an asymptomatic carrier was considered to be a healthy individual with the presence of anti-leishmanial antibodies and/or a positive rt-PCR in blood. These patients correspond to the study published by Aliaga et al. [6].

### 4.4. Leishmania Disease

The Leishmania disease (LD) group was composed of 26 individuals (18 VL, 6 CL, and 2 ML). The average age of the group is 31 (range 1–78 years), with 34.8% of its members being women (*n* = 9).

The diagnosis was made by commercial rt-PCR from peripheral blood or bone marrow samples in cases of visceral leishmaniasis. In cutaneous and mucosal leishmaniasis, rt-PCR was performed from skin and mucosal biopsy, respectively [36]. In addition, microscopic visualization was performed in biopsy and bone marrow samples.

### 4.5. Infected Group

This cohort is the result of the union of the two previous cohorts (asymptomatic and LD). It includes 81 individuals, with a mean age of 38 years (range 1 to 78 years), and 46.9% are women (*n* = 38). This group refers to people infected with Leishmania, independently of their symptomatology.

### 4.6. Cohabiting Group

The cohabiting group (*n* = 32) was made up of healthy people living with patients with Leishmania disease. Serology and rt-PCR in these patients were negative. The average age of the group is 47 (range 6 to 61 years), with 56% of its members being women (*n* = 18).

### 4.7. Blood Collection

Two peripheral blood samples were collected from each study participant for serological and molecular studies: one in a tube without an anticoagulant containing 8.5 cubic centimeters of blood and another in a tube with EDTA containing 3.5 cubic centimeters of blood.

### 4.8. Serological Test

A commercial indirect fluorescent antibody technique (IFA) (Leishmania IFA IgG, Vircell, Granada, Spain) was used for the detection of antibodies against *L. infantum*. IFA assay was carried out following the manufacturer’s protocols. A single determination was performed for each serum sample, and we considered a positive result when the IgG antibody titer was ≥1:80. To carry out the test, 25 µL of the appropriate dilution of each sample were deposited onto slides containing the antigens, and the slides were incubated (37 °C for 30 min), washed with PBS, and then dried in a humidity chamber. Then, 25 µL of fluorescein isothiocyanate (FITC)-anti-IgG antibody was added to the concavities. The slides were incubated (37 °C for 30 min), washed, and read using a fluorescence microscope with 400× magnification. The reading was performed by two independent microscopists.

### 4.9. DNA Extraction

The DNA was obtained using the kit extracted using the QIAMP DNA Mini Kit (Qiagen, Hilden, Germany) according to the manufacturer’s instructions. The extracted DNA was kept at −20 °C until its amplification by PCR.

### 4.10. Leishmania PCR

A commercial rt-PCR (Realcycler Leishmania Donovani Complex, Progenie, Spain) has been used for the qualitative detection of nucleic acids from Leishmania sp. The assay was carried out following the manufacturers’ instructions, carrying out the amplification and detection reaction in a CFX thermocycler (CFX96 Touch Real-Time PCR Detection System, Bio-Rad, Hercules, CA, USA).

### 4.11. HLA Class I and II Genotyping

High-resolution genotyping of HLA class I (A, B, and C) and II (DRB1 and DQB1) loci was performed using the LABType sequence-specific oligonucleotide typing test (One Lambda, Canoga Park, CA, USA). Target DNA was amplified by PCR using sequence-specific primers, followed by hybridization with allele-specific oligodeoxynucleotides coupled with fluorescent phycoerythrin-labelled microspheres. Fluorescence intensity was determined using a LABScan 100 system (Luminex xMAP, Austin, TX, USA). HLA alleles were assigned using the HLA-Fusion software, version 4.6.0 (Palex Medical, Barcelona, Spain). However, the low resolution was used to reduce the number of comparisons to be made and increase the statistical power of the analysis.

### 4.12. Haplotypes, Bw4/Bw6 Epitopes and C1/C2 Groups Analysis

For the haplotype study, we extracted the haplotypes of each studied participant using the extended haplotypes published by MT Dorak et al. [37]. Subsequently, we made comparisons between the groups.

To study the Bw4/Bw6 epitopes, we determined the presence of these epitopes in each HLA allele and applied them to each participant [38]. Subjects were classified into homozygotes or heterozygotes based on the presence of one or both epitopes. Subsequently, we performed comparisons between groups.

Finally, the analysis of the C1 and C2 groups was performed by classifying the HLA-C alleles in each group [15]. This was applied to each subject, and they were classified into homozygotes or heterozygotes based on the C1/C2 groups. Subsequently, we performed comparisons between groups.

### 4.13. Statistical Analysis

The Shapiro–Wilk test was applied to evaluate the normality of the quantitative variables of interest in each group. The Kruskal–Wallis test was used to compare the quantitative variable between more than two independent groups. Frequencies of individual HLA alleles in patients and controls were compared using the χ2-test. Variants with expected counts less than five were combined into a common class (binned) before computing the χ2-test. Significance levels were corrected by Bonferroni correction for a multiplicity of testing by the number of comparisons. The Bonferroni correction is used in HLA statistical studies to address the problem of multiple comparisons. When multiple statistical tests are performed simultaneously, the probability of obtaining significant results simply by chance increases. A corrected *p* value of <0.05 was considered statistically significant for all statistical tests. The degree of association between the response and independent variables was determined by calculating the crude Odds Ratio (cOR) and its 95% confidence interval (CI). To determine the independent effect of each factor, the OR was adjusted (aOR) using a binomial logistic regression model. Adjustment was made for age and sex. All the analyses were performed using SPSS statistical software (Windows version 26, IBM, Armonk, NY, USA).

## Figures and Tables

**Table 1 ijms-25-08205-t001:** Comparison between different cohorts of HLA class I and II alleles with *p* value ≤0.05 before Bonferroni adjustment.

	LD (*n* = 26)	Cohabitants (*n* = 32)	Asymptomatic (*n* = 55)	Infected (*n* = 81)	Controls (*n* = 636)	P1(Pc)	P2(Pc)	P3(Pc)
HLA-A*24	0.173	0.172	0.164	0.167	0.108	0.029 (n.s)	n.s	n.s
HLA-A*34	0.038	0	0	0.012	0.050	n.s	0.020 (n.s)	n.s
HLA-B*07	0.154	0.203	0.100	0.117	0.097	n.s	n.s	0.007 (n.s)
HLA-B*15	0.058	0.125	0.100	0.086	0.052	n.s	n.s	0.013 (n.s)
HLA-B*38	0.019	0.094	0.055	0.043	0.025	n.s	n.s	**0.001 (0.026)**
HLA-B*44	0.115	0.063	0.127	0.123	0.154	n.s	n.s	0.045 (n.s)
HLA-C*03	0.077	0.173	0.091	0.086	0.045	n.s	n.s	**6 × 10^−6^ (8.4 × 10^−5^)**
HLA-DR*11	0.135	0.234	0.164	0.167	0.108	n.s	n.s	0.007 (n.s)

LD: Leishmania disease; P1: Infected vs. Controls; P2: Leishmania disease vs. Controls; P3: Cohabitants vs. Controls; Pc: P corrected by Bonferroni; n.s: not significant. Significant values after the Bonferroni correction are in bold.

**Table 2 ijms-25-08205-t002:** Comparison between different cohorts of haplotypes with *p* value ≤0.05 before Bonferroni adjustment.

Haplotype	LD (*n* = 26)	Cohabitants (*n* = 32)	Controls (*n* = 636)	P1(Pc)	P2(Pc)
18.3	0.038	0.031	0.004	0.030 (n.s)	n.s
60.3	0.019	0.031	0.003	n.s	0.028 (n.s)

LD: Leishmania disease; P1: Leishmania disease vs. Controls; P2: Cohabitants vs. Controls; Pc: P corrected by Bonferroni; n.s: not significant.

**Table 3 ijms-25-08205-t003:** Frequency of homozygosity at each HLA loci and cohort.

	Homozygosis HLA-A	Homozygosis HLA-B	Homozygosis HLA-C	Homozygosis HLA-DRB1	Homozygosis HLA-DQB1
LD (*n* = 26)	0.154	0.154	0.192	0.115	0.269
Cohabitans (*n* = 32)	0.188	0.125	0.063	0.125	0.313
Asymptomatic (*n* = 55)	0.200	0.109	0.127	0.091	0.200
Infected (*n* = 81)	0.185	0.123	0.148	0.099	0.222
Controls (*n* = 636)	0.126	0.085	0.127	0.132	0.253

LD: Leishmania disease.

**Table 4 ijms-25-08205-t004:** Frequency of heterozygotes and homozygotes of Bw4/Bw6 epitopes, and C1/C2 groups.

	Heterozygous Bw4/Bw6	Homozygous Bw4	Homozygous Bw6	Heterozygous C1/C2	Homozygous C1	Homozygous C2
LD (*n* = 26)	0.654	0.154	0.192	0.462	0.308	0.231
Cohabitans (*n* = 32)	0.563	0.156	0.281	0.469	0.438	0.094
Asymptomatic (*n* = 55)	0.618	0.145	0.236	0.491	0.382	0.127
Infected (*n* = 81)	0.630	0.148	0.222	0.481	0.358	0.160
Controls (*n* = 636)	0.629	0.189	0.182	0.476	0.316	0.208

LD: Leishmania disease.

**Table 5 ijms-25-08205-t005:** The demographics of the different study groups.

	LD	Cohabitants	Asymptomatic	Infected	Controls
*n*	26	32	55	81	636
Females (%)	9 (34.8%)	18 (56%)	23 (41.8)	38 (46.9%)	324 (51%)
Age (range)	31 (1–78)	47 (6–61)	41 (18–65)	38 (1–78)	45 (18–65)

LD: Leishmania disease.

## Data Availability

The data of this study will be provided upon requests sent to the authors/corresponding author.

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
