# Peer review of "Analysis of HLA Alleles in Different Cohorts of Patients Infected by L. infantum from Southern Spain"

_ijms, 2024, doi:10.3390/ijms25158205_

Round 1

Reviewer 1 Report

Comments and Suggestions for Authors

General Remarks: The study on the association between Leishmania infantum and HLA alleles tackles a critical public health concern, especially in regions where the disease is endemic. This research is both relevant and timely, offering valuable insights into disease progression and potential therapeutic approaches.

However, revisions are necessary in several sections of the manuscript. Here are a few recommendations:  the introduction needs a more focused narrative, the discussion section requires a more robust analysis and interpretation of the findings, and the materials and methods section should be more detailed and precise.

Introduction Section:

1.     Ensure all numerical claims and facts are properly cited. For example,

Line 40 Mortality………in the poorest country. Is properly cited?????????

2.     The authors should consider ending the introduction with a sentence that outlines the main objectives of the study. For example, The main objectives of this study were to analyse the association of HLA alleles with susceptibility and progression of L. infantum infection, and to identify potential protective or risk HLA alleles.

Discussion Section:

1. Leishmaniasis, including infection caused by Leishmania infantum, is recognised as a significant public health problem by the World Health Organization (WHO). Including the WHO's efforts and recommendations can provide valuable context to the discussion section of the paper. The WHO emphasises the importance of understanding host-pathogen interactions and the role of genetic factors in disease susceptibility and resistance. https://www.who.int/health-topics/leishmaniasis#tab=tab_1

Materials and Methods:

1. The authors should provide a brief explanation of why Bonferroni correction was used, it would enhance clarity.

Author Response

Dear Reviewer,

Thank you for your detailed comments and suggestions on our manuscript entitled “Analysis of HLA alleles in different cohorts of patients infected by L. infantum  from the Southern of Spain”. Your comments have helped us to identify areas for improvement and to strengthen our work.

We appreciate the time and effort spent on the review and have carefully considered your recommendations. Below, we respond to each of your comments and detail the modifications made to the manuscript.

Introduction Section:

  1. Ensure all numerical claims and facts are properly cited. For example, Line 40 Mortality………in the poorest country. Is properly cited?????????

Thank you for reporting this error. We have checked the text references. We hope they are correct.

  1. The authors should consider ending the introduction with a sentence that outlines the main objectives of the study. For example, The main objectives of this study were to analyse the association of HLA alleles with susceptibility and progression of L. infantum infection, and to identify potential protective or risk HLA alleles.

We have added the sentence you suggest at the end of the introduction (Line 94).

Discussion Section:

  1. Leishmaniasis, including infection caused by Leishmania infantum, is recognised as a significant public health problem by the World Health Organization (WHO). Including the WHO's efforts and recommendations can provide valuable context to the discussion section of the paper. The WHO emphasises the importance of understanding host-pathogen interactions and the role of genetic factors in disease susceptibility and resistance. https://www.who.int/health-topics/leishmaniasis#tab=tab_1

We have added in the discussion section the efforts and importance of leishmaniasis for WHO (line 176). We believe this improves the introduction of the discussion. We would like to thank the reviewer for the suggestion.

Materials and Methods:

  1. The authors should provide a brief explanation of why Bonferroni correction was used, it would enhance clarity.

We expand the materials and methods section by explaining why Bonferroni correction is used in HLA comparisons (line 355).

Once again, we sincerely thank you for your valuable contributions to our manuscript. Your comments have been of great help in improving the quality of our work. We hope that the modifications made will meet your expectations and remain at your disposal for any further clarification.

Kind regards,

Dr. Juan Francisco Gutiérrez-Bautista

Servicio de Análisis Clínicos e Inmunología

University Hospital Virgen de las Nieves.

[email protected]

Reviewer 2 Report

Comments and Suggestions for Authors

In the present manuscript, Gutiérrez-Bautista and colleagues investigated the distribution of HLA alleles in different L. infantum infected (or exposed to infection) and control groups. Their results show that HLA-B*38 and HLA-C*03 alleles are strongly associated with protection against infection.

Comments and suggestions:

-          The study groups are introduced in Materials and Methods. It is recommended that the groups are described in tabular form (age, gender composition, etc.) and that statistical comparisons are made at the beginning of the results section. This is also suggested because differences in sex ratio or age between different groups may affect the results, as these factors are related to the individual immune response.

-          The tables are not properly titled and do not give the reader enough information to understand them. I also recommend that abbreviations are footnoted. Please also indicate the sample number (n).

-          It is not clear from the text of the manuscript what the corrected p-value was after the Bonferroni correction. Please indicate this in the tables.

-          For the sub-chapter "Haplotypes and Bw4/Bw6, C1/C2 epitopes analysis", it is not sufficient to simply refer to the analyses carried out by references, but at least a brief description of the methods is required. The same applies to the other sub-chapters to help the reader understand the results and methods.

-          The analysis of the data is limited to an overall comparison between frequencies, but the sample size would allow more complex analyses to be carried out. I also recommend performing logistic regression (if possible ROC curve analysis). Adjust for age, sex, and other available covariates (if any) in statistical analyses.

The usefulness and validity of the results, which are the subject of this manuscript, are limited by the simplicity of the statistical methods used. I recommend that more complex analyses be carried out and the manuscript revised on the basis of the results.

Comments on the Quality of English Language

The quality of the English is fair, but I recommend a thorough revision of the manuscript.

Author Response

Dear reviewer,

Thank you for your constructive criticisms and recommendations on our manuscript entitled “Analysis of HLA alleles in different cohorts of patients infected by L. infantum from the Southern of Spain”. Your comments have provided us with valuable insight to improve and refine our work.

We have revised and adjusted the manuscript based on your suggestions. Below, you will find our responses to each of your comments, along with the modifications made.

Comments and suggestions:

-          The study groups are introduced in Materials and Methods. It is recommended that the groups are described in tabular form (age, gender composition, etc.) and that statistical comparisons are made at the beginning of the results section. This is also suggested because differences in sex ratio or age between different groups may affect the results, as these factors are related to the individual immune response.

We have added a new table (Table 5) showing the demographic characteristics of the groups studied (line 268). We have also included at the beginning of the results section the statistical analysis comparing age and sex between the groups. We thank the reviewer for this valuable comment.

-          The tables are not properly titled and do not give the reader enough information to understand them. I also recommend that abbreviations are footnoted. Please also indicate the sample number (n).

We have improved the title of the tables. We revised the table footnoted to improve them. We have added the sample number of each group.

-          It is not clear from the text of the manuscript what the corrected p-value was after the Bonferroni correction. Please indicate this in the tables.

The Bonferroni-corrected p-value is shown in the tables in parentheses. It corresponds to Pc, which is explained in the table footnote. In addition, The Bonferroni-corrected p-value is shown in the tables in parentheses. It corresponds to Pc, which is explained in the table footnote. In addition, significant values after the Bonferroni correction are in bold.

-          For the sub-chapter "Haplotypes and Bw4/Bw6, C1/C2 epitopes analysis", it is not sufficient to simply refer to the analyses carried out by references, but at least a brief description of the methods is required. The same applies to the other sub-chapters to help the reader understand the results and methods.

Thank you for this suggestion. We have improved the explanation of the procedure to study haplotypes, Bw4/Bw6 epitopes, and C1/C2 groups (line 336).

-          The analysis of the data is limited to an overall comparison between frequencies, but the sample size would allow more complex analyses to be carried out. I also recommend performing logistic regression (if possible ROC curve analysis). Adjust for age, sex, and other available covariates (if any) in statistical analyses.

We performed logistic regression analysis to evaluate the associations between our variables of interest and the presence of infection.

In this analysis, we adjusted for available covariates, specifically age and sex, to control for potential confounders.

We also performed ROC curve analysis to assess the predictive ability of our model. However, the results of the ROC curves did not show any significant values indicating a strong discriminative ability of the model. Therefore, we did not include it in the study.

Again, thank you for your detailed review and valuable suggestions. We believe that the improvements implemented strengthen our work and trust that it will now meet your expectations. We remain at your disposal for any further questions.

Kings regards,

Dr. Juan Francisco Gutiérrez-Bautista.

Servicio de Análisis Clínicos e Inmunología

University Hospital Virgen de las Nieves.

[email protected]

Reviewer 3 Report

Comments and Suggestions for Authors

The manuscript "Analysis of HLA alleles in different cohorts of patients infected by L. infantum from the Southern of Spain" explores the association between specific HLA variants and Leishmania infantum infection in a southern Spain cohort. Leishmaniasis is considered as a re-emerging infection based on the geographic region and the outcome of leishmanias is known to depend vastly on Leishmania-host interaction. The submitted manuscript identified a possible association of HLA-B*38 and HLA-C*03 alleles with protection against infection in the southern Spain cohort. This expands the geographical location specific association of HLA alleles in Leishmaniasis.

After going through the manuscript, I have following comments for the authors.

1.     Previously. protection-associated HLA alleles have been known to display common epitope binding motifs that maps to amino acid substitutions shared across these alleles. Was this aspect analyzed in the study?

2.     Please briefly discuss the therepeutic significance of the knowledge related to the association between specific HLA variants and Leishmaniasis.

3.     What was the house-keeping gene used in rt-PCR analysis?

Comments on the Quality of English Language

Minor grammatical corrections and few syntax adjustments recommended.

Author Response

Dear Reviewer,

Thank you for your detailed comments and suggestions on our manuscript entitled “Analysis of HLA alleles in different cohorts of patients infected by L. infantum from the Southern of Spain.” Your insights have been invaluable in identifying areas for improvement and enhancing the quality of our work.

We appreciate the time and effort you have dedicated to reviewing our manuscript. We have carefully considered your recommendations and have made the necessary modifications. Below, we respond to each of your comments and detail the changes made to the manuscript.

  1. Previously. protection-associated HLA alleles have been known to display common epitope binding motifs that maps to amino acid substitutions shared across these alleles. Was this aspect analyzed in the study?

We have not performed this type of analysis.

  1. Please briefly discuss the therepeutic significance of the knowledge related to the association between specific HLA variants and Leishmaniasis.

Added in discussion (line 183).

  1. What was the house-keeping gene used in rt-PCR analysis?

We have not added the rt-PCR target because it is a commercial kit that does not specify the target.

We extend our sincere gratitude for your invaluable contributions to our manuscript. Your feedback has significantly enhanced the quality of our work. We trust that the revisions we've implemented meet your expectations, and we are available to address any additional questions or concerns you may have.

Kind regards,

Dr. Juan Francisco Gutiérrez-Bautista

Servicio de Análisis Clínicos e Inmunología

University Hospital Virgen de las Nieves.

[email protected]

Reviewer 4 Report

Comments and Suggestions for Authors

The submitted manuscript presents the results of a study exploring the association between specific HLA variants and Leishmania infantum infection. The study indicates that the HLA-B*38 and HLA-C*03 alleles may be associated with protection against L. infantum infection. Hence, the study adds to the existing body of evidence on the role of HLA in leishmaniasis, which may potentially be translated into new therapeutic approaches.

The manuscript exhibits a logical structure, and the text is suitably complemented by several  tables and supplementary files. The methodological approach and statistical analyses employed in the study are sufficiently detailed in the text. I have following comments and suggestions regarding some aspects of the manuscript or the study:

1.) In Materials and Methods, you state that HLA typing was done at a high-resolution level. However, the Results section as well as Table 1 and Supplementary Tables 1 and 2 present the allele frequencies only at the low-resolution level. Can you explain the reason for this decision?

2.) In the Discussion, it would be also appropriate to discuss the limitations of the study, most importantly the small sample sizes of asymptomatic infection, LD, and cohabiting groups, which likely result into small statistical power of the study.

3.) Readers unfamiliar with the HLA system would certainly appreciate a brief explanation in the Introduction chapter of what the Bw4, Bw6, C1 and C2 epitopes/groups are, and why you decided to analyze them in the context of the risk/protection from L. infantum infection. This information is partially provided only later in the Discussion.

4.) In several places in the manuscript you refer to C1 and C2 as groups, while in other places as epitopes. What is the correct form?

5.) In Supplementary Table 3, you could indicate the exact allelic composition of individual haplotypes, as you do in the text of chapter 2.2 for haplotypes 18.3 and 60.3.

6.) In some places of the text, you are confusing genes with molecules. For example: "These molecules belong to the major histocompatibility complex (MHC) and are the most polymorphic molecules in the human genome."

Comments on the Quality of English Language

The stylistic, linguistic and formal treatment of the manuscript is at a sufficiently good level. However, there are some grammatical, stylistic or linguistic errors as well as unclear or incorrect formulations in the text which require attention. Here are some examples:

Introduction: "In the search for parameters that may explain the development of symptomatic or asymptomatic disease, as well as risk or protection from infection, some several genes and polymorphisms are associated with leishmaniasis."

Introduction: "We performed molecular HLA typing of the individuals included in each group and analyzed the allelic and haplotypic frequencies, homozygosity, and the Bw4, Bw6, C1, and C2 epitopes about to the development of symptomatic or asymptomatic infection, as well as the protective or risk role that HLA molecules may play."

Results: "Regarding the HLA-II alleles, only the HLA-DRB1*11 allele was close to significance, not supporting the Bonferroni correction."

Results: "However, they did not support Bonferroni´s correction."

Discussion: "Therefore, we performed the analysis of allelic and haplotypic frequencies of HLA genes to determine associations of susceptibility or protection."

Discussion: "However, only HLA-B*38 and HLA-C*03 alleles supported correction by multiple testing."

Discussion: " In our results, despite not supporting the Bonferroni correction, it tends to be associated with a protective role ..."

Discussion: "Most of our results indicate a predominant role of HLA-I versus HLA-II molecules about L. infantum infection."

Discussion: "HLA-I molecules are responsible for the antigenic presentation of intracellular proteins; thus, it is responsible for presenting peptides derived from intracellular pathogens."

Discussion: This haplotype contains the HLA-C*03 allele, which as previously mentioned is associated with protection in our cohort, there being no previously published results on this allele."

Discussion: "... of AISD in HIV patients".

Author Response

Dear Reviewer,

Thank you for your constructive criticisms and recommendations on our manuscript entitled “Analysis of HLA alleles in different cohorts of patients infected by L. infantum from the Southern of Spain”. Your comments have provided us with valuable insight to improve and refine our work.

We have revised and adjusted the manuscript based on your suggestions. Below, you will find our responses to each of your comments, along with the modifications made.

1.) In Materials and Methods, you state that HLA typing was done at a high-resolution level. However, the Results section as well as Table 1 and Supplementary Tables 1 and 2 present the allele frequencies only at the low-resolution level. Can you explain the reason for this decision?

We have added the explanation in the materials and methods section (line 333). We used low resolution due to the low number of cases, to decrease the number of alleles and increase statistical power.

2.) In the Discussion, it would be also appropriate to discuss the limitations of the study, most importantly the small sample sizes of asymptomatic infection, LD, and cohabiting groups, which likely result into small statistical power of the study.

Added at the end of the discussion (line 257).

3.) Readers unfamiliar with the HLA system would certainly appreciate a brief explanation in the Introduction chapter of what the Bw4, Bw6, C1 and C2 epitopes/groups are, and why you decided to analyze them in the context of the risk/protection from L. infantum infection. This information is partially provided only later in the Discussion.

We have extended the introduction with an explanation of the Bw4/Bw6 epitopes and the C1/C2 groups (line 72). We thank the reviewer for this suggestion.

4.) In several places in the manuscript you refer to C1 and C2 as groups, while in other places as epitopes. What is the correct form?

The correct way is to talk about groups C1 and C2. We have corrected the errors throughout the text.

5.) In Supplementary Table 3, you could indicate the exact allelic composition of individual haplotypes, as you do in the text of chapter 2.2 for haplotypes 18.3 and 60.3.

Added in Supplementary Table 3.

6.) In some places of the text, you are confusing genes with molecules. For example: "These molecules belong to the major histocompatibility complex (MHC) and are the most polymorphic molecules in the human genome."

We have reviewed the entire text and corrected the errors (line 59).

Comments on the Quality of English Language

The stylistic, linguistic and formal treatment of the manuscript is at a sufficiently good level. However, there are some grammatical, stylistic or linguistic errors as well as unclear or incorrect formulations in the text which require attention. Here are some examples:

Introduction: "In the search for parameters that may explain the development of symptomatic or asymptomatic disease, as well as risk or protection from infection, some several genes and polymorphisms are associated with leishmaniasis."

Changed in the manuscript (line 54).

Introduction: "We performed molecular HLA typing of the individuals included in each group and analyzed the allelic and haplotypic frequencies, homozygosity, and the Bw4, Bw6, C1, and C2 epitopes about to the development of symptomatic or asymptomatic infection, as well as the protective or risk role that HLA molecules may play."

Changed in the manuscript (line 90).

Results: "Regarding the HLA-II alleles, only the HLA-DRB1*11 allele was close to significance, not supporting the Bonferroni correction."

Changed in the manuscript.

Results: "However, they did not support Bonferroni´s correction."

Changed in the manuscript (line 150).

Discussion: "Therefore, we performed the analysis of allelic and haplotypic frequencies of HLA genes to determine associations of susceptibility or protection."

Changed in the manuscript (line 183).

Discussion: "However, only HLA-B*38 and HLA-C*03 alleles supported correction by multiple testing."

Changed in the manuscript (line 188).

Discussion: " In our results, despite not supporting the Bonferroni correction, it tends to be associated with a protective role ..."

Changed in the manuscript (line 199).

Discussion: "Most of our results indicate a predominant role of HLA-I versus HLA-II molecules about L. infantum infection."

Changed in the manuscript (line 215).

Discussion: "HLA-I molecules are responsible for the antigenic presentation of intracellular proteins; thus, it is responsible for presenting peptides derived from intracellular pathogens."

Changed in the manuscript (line 216).

Discussion: This haplotype contains the HLA-C*03 allele, which as previously mentioned is associated with protection in our cohort, there being no previously published results on this allele."

Changed in the manuscript (line 227).

Discussion: "... of AISD in HIV patients".

Changed in the manuscript (line 249).

We thank you again for your time and constructive comments. We believe that the revisions have significantly improved our manuscript and hope that these modifications are satisfactory. We look forward to any other suggestions or questions you may have.

King regards,

Dr. Juan Francisco Gutiérrez-Bautista

Servicio de Análisis Clínicos e Inmunología

University Hospital Virgen de las Nieves.

[email protected]

Round 2

Reviewer 2 Report

Comments and Suggestions for Authors

I accept the Authors' answers to my questions and comments.